# IgFlow-LM: De Novo Antibody Design via Joint Flow Matching on SE(3) and Protein Language Models Probability Flows

## Abstract

In this work, we present IgFlow-LM, a multi-modal deep generative model for de novo antibody design based on a flow-matching framework that integrates protein language models (PLMs). By learning the joint distribution over SE(3)-equivariant structural flows and PLM-derived probabilistic flows, IgFlow-LM enables the coordinated generation of antibody 3D structures and latent embeddings in the PLM space. Experimental results demonstrate that, in unconditional design, IgFlow-LM generates antibody structures that closely resemble naturally occurring antibodies. IgFlow-LM generates antibodies closely resembling naturally observed ones, with backbone dihedral angles exhibiting strong agreement with reference antibody distributions and overall backbone conformations adhering more closely to physical constraints. Furthermore, we benchmark IgFlow-LM against baseline models on two commonly studied conditional CDR design tasks. IgFlow-LM demonstrates superior overall performance compared to baselines and generates CDR sequences with higher diversity.

## 1 Introduction

In the field of protein engineering, an important application is the design of immunoglobulins (antibodies) (Shanehsazzadeh et al., 2023; Ruffolo et al., 2024). Immunoglobulins play a central role in assisting the adaptive immune system to recognize and neutralize pathogens (Raybould et al., 2019; Kong et al., 2023a). They are composed of two heavy chains and two light chains, each of which consists of both constant and variable domains (Schroeder Jr & Cavacini, 2010). The variable regions contain highly diverse loops known as complementarity-determining regions (CDRs), which are critical for antigen recognition and binding (Narciso et al., 2011). Among these, the third CDR of the heavy chain (HCDR3) exhibits the greatest sequence variability and is the most actively involved in antigen binding, often serving as the primary determinant of antigen specificity (Tsuchiya & Mizuguchi, 2016). Consequently, antibody design efforts frequently focus on the conditional design of CDRs, with particular emphasis on accurately determining the HCDR3 region.

Generative models have recently gained significant prominence in protein engineering (Ingraham et al., 2019; Kucera et al., 2022; Alford et al., 2017). Given that antibody function is largely determined by its sequence (Whisstock & Lesk, 2003; Robinson, 2015), simultaneous optimization of both sequence and structure during generation is essential to enhance their consistency. Protein language models (PLMs) have been shown to capture a range of biologically relevant features, including structural (Rao et al., 2020; Lin et al., 2023) and functional properties (Nijkamp et al., 2023). By jointly modeling PLM latent space embeddings and antibody structures, generative models can leverage the evolutionary information encoded within PLMs to improve the foldability and functionality of generated sequences, while also enabling more effective exploration of the antibody structural landscape—particularly for tasks involving highly diverse regions such as complementarity-determining region (CDR) loops. This approach offers a dual advantage: It avoids the error propagation associated with traditional fixed backbone sequence design methods (Dreyer et al., 2023; Lu et al., 2024) and avoids the reduction in sequence diversity often caused by joint modeling with discrete modalities (Campbell et al., 2024; Nagaraj et al., 2024).

Motivated by this, we propose a multimodal deep generative model for antibody design, named IgFlow-LM. This model is built upon a conditional flow matching (CFM) framework that enables the joint generation of PLM latent embeddings and antibody structures. Specifically, we extend the flow-matching-based protein design model FrameFlow (Yim et al., 2023a) to simultaneously generate PLM latent embeddings and antibody structures, and train the model on the Structural Antibody Database (SAbDab) (Dunbar et al., 2014). For the choice of PLM, we adopt IgBert (Kenlay et al., 2024), which is specifically trained on the Observed Antibody Space (OAS) (Olsen et al., 2022), as the source of antibody sequence embeddings. IgBert is capable of taking both heavy and light chain sequences as input and producing their corresponding latent space representations. Our focus is on generating novel antibody variable domains, and we evaluate the performance of our model in the following tasks: (1) unconditional generation of paired heavy and light chains; and (2) framework-conditioned loop design of CDRs. In summary, the main contributions of this work are as follows:

- We propose IgFlow-LM, a multi-modal deep generative model for de novo antibody design that integrates PLM latent embeddings within a flow matching framework.

- We modify the architecture of the base FrameFlow model to enable joint generation of antibody structures and PLM latent embeddings, and further extend it to the task of framework-conditioned CDR loop design.

- Experimental results demonstrate that IgFlow-LM generates antibodies that better conform to physical constraints compared to baseline models, showcasing its effectiveness in antibody engineering applications.

## 2    RELATED WORKS

Diffusion models progressively transform prior samples into meaningful outputs through a step-by-step denoising process (Ho et al., 2020; Vincent, 2011; Sohl-Dickstein et al., 2015), and have been widely applied in image (Ho et al., 2020; Zhang et al., 2024), text (Li et al., 2022; Hoogeboom et al., 2021), and molecular generation (Watson et al., 2023; Lisanza et al., 2024; Hoogeboom et al., 2022; Jing et al., 2020). Recently, continuous normalizing flows based on ordinary differential equations (ODEs) (Pooladian et al., 2023; Chen et al., 2018) have emerged as a promising alternative to diffusion models. In particular, CFM (Lipman et al., 2022) directly learns the ODE-based probabilistic path from a prior distribution to a target distribution, avoiding the computationally expensive simulation procedures required by diffusion models (Lipman et al., 2022; Song et al., 2020a; Chen & Lipman, 2023; Ben-Hamu et al., 2022; De Bortoli et al., 2022).

Generative models have demonstrated remarkable performance in protein-related applications (Watson et al., 2023; Yim et al., 2023a; Jain et al., 2022; Khan et al., 2023; Chen et al., 2024; Trippe et al., 2022; Luo et al., 2025). Current antibody design approaches can be broadly categorized into three classes: sequence design (Dreyer et al., 2023; Hie et al., 2024), structure design (Cutting et al., 2025; Bennett et al., 2023), and sequence-structure co-design (Nagaraj et al., 2024; Jin et al., 2021; Kong et al., 2023b). Sequence-structure co-design has become increasingly prevalent in recent antibody modeling efforts. However, existing methods for generating antibody sequence-structure pairs typically operate directly on discrete sequences, an approach that may be suboptimal for preserving structural diversity. Such discretization can lead to premature convergence toward commonly observed sequence-structure motifs, thereby compromising the exploration of diverse solutions (Nagaraj et al., 2024).

## 3    METHODOLOGY

In this section, we first review the concept of the CFM framework. Subsequently, we describe the extension of FrameFlow to antibody design, enabling the joint generation of two modalities: antibody structures and PLMs latent embeddings. Figure 1 provides an overview of IgFlow-LM, illustrating the architecture and workflow of the proposed multi-modal generative framework. The code for IgFlow-LM is available at https://anonymous.4open.science/r/IgFlow-LM-8848/.

Figure 1: Overview of the IgFlow-LM architecture. IgFlow-LM takes as input initial noise-perturbed antibody structures and noisy PLM latent embeddings. An encoder first processes these inputs together with the current time step to produce initial frames $\mathbf{T}^0$, node embeddings $\mathbf{h}^0$, and edge embeddings $\mathbf{z}^0$. Invariant Point Attention (IPA) updates the node features by integrating information from the explicit residue frames $\mathbf{T}^l$, as well as the PLM embedding and timestep information encoded in the input embeddings. The updated features are added to the input node features and subsequently fed into a Transformer block to capture inter-residue dependencies. Subsequently, the output is summed with the embeddings from the IPA module and fed into a multi-layer perceptron (MLP) to obtain the updated node embeddings $\mathbf{h}^{l+1}$. Additionally, the updated node embeddings are used to refine the edge embeddings $\mathbf{z}^l$ and the frame representations $\mathbf{T}^l$. Through multiple steps of ODE integration, the model generates a novel antibody sequence-structure pairs.

## 3.1 SE(3)-Flow Matching for Antibody Structure Generation

In this study, we employ SE(3)-flow matching to model the generative process of continuous backbone frameworks. The backbone atomic coordinates of the i-th residue in an antibody scaffold are parameterized by a rigid transformation $\mathrm{T}^i \in \mathrm{SE}(3)$ and three fixed heavy atoms $N^*, C^*, C^*_\alpha \in \mathbb{R}^3$ (Jumper et al., 2021; Köhler et al., 2020):

$$[N^i, C^i, C^i_\alpha] = T^i \cdot [N^*, C^*, C^*_\alpha] \tag{1}$$

where $\mathrm{T}^i$ is an element of the special Euclidean group (the set of valid translations and rotations in Euclidean space). That is, the backbone of an antibody of length L can be parameterized as $\mathbf{T} = [T^{(1)}, ..., T^{(L)}]$, with $\mathbf{T} \in \mathrm{SE}(3)^L$. Each frame can be decomposed as $T^i = (r^i, x^i)$, where $x^i \in \mathbb{R}^3$ represents the translation vector corresponding to the $\mathrm{C}^i_\alpha$ atom, and $r^i \in \mathrm{SO}(3)$ is a 3×3 rotation matrix, which can be derived from relative atomic positions via the Gram-Schmidt process. Following the framework of FrameFlow (Yim et al., 2023a), we initialize the distribution $p_0(\mathbf{T}_0) = p_{\mathrm{noise}}(\mathbf{T}_0)$ and learn a time-dependent vector field $v_\theta(\mathbf{T}_t, t)$ to evolve these frames. This vector field describes a structure flow ordinary differential equation with initial condition $\mathbf{T}_0 \sim p_0$:

$$\frac{d\mathbf{T}_t}{dt} = v_\theta(\mathbf{T}_t, t) \tag{2}$$

The target distribution is defined as the distribution of antibody backbone structures $p_1(\mathbf{T}_1)$. We approximate the SE(3)-equivariant flow by fitting a conditional vector field $u_t(\mathbf{T}_t|\mathbf{T}_1, \mathbf{T}_0)$ using a neural network. This conditional vector field represents an interpolation between the initial (noisy) frames and the final target frames through a series of probability distributions. In other words, the objective loss aims to minimize the discrepancy between the predicted final target and the true target, given the noisy initial condition:

$$\mathcal{L}_{\mathrm{SE}(3)} = \mathbb{E}_{t, p_1(\mathbf{T}_1), p_0(\mathbf{T}_0)}[||u_t(\mathbf{T}_t|\mathbf{T}_1, \mathbf{T}_0) - v_\theta(\mathbf{T}_t, t)||^2] \tag{3}$$

After training, we can perform flexible sampling of antibody structures by reverse integrating the ODE with a specified number of time steps and a variance schedule. We provide a more complete description in the Appendix.

## 3.2 PLM-based Probability Flows Matching for PLMs Embeddings

Since the latent embeddings provided by PLMs are continuous variables, they are inherently more suitable for continuous FM-based methods (Hu et al., 2024). To leverage the semantically meaning-

ful latent space offered by PLMs, we can directly define the mapping between the initial distribution $p_0 = p_{\text{noise}}(\mathbf{x}_0)$ and the target distribution $p_1 = p(\mathbf{x})$ as a time-dependent vector field:

$$\frac{d\mathbf{x}_t}{dt} = v_\theta(\mathbf{x}_t, t) \tag{4}$$

This time-dependent vector field describes an ODE with an initial condition $\mathbf{x}_0 \sim p_0$. The target distribution corresponds to the distribution of PLM latent embeddings $p_1(\mathbf{x}_1)$. Similar to the SE(3)-equivariant flow, we approximate this PLM embedding flow using a conditional vector field $u_t(\mathbf{x}_t|\mathbf{x}_1, \mathbf{x}_0)$, which is modeled by a neural network. This conditional vector field represents an interpolation between the initial (noisy) embedding and the final target embedding through a series of probability distributions. A more detailed description of this process is provided in the Appendix. As with the SE(3)-equivariant flow, the objective loss aims to minimize the discrepancy between the predicted final target and the true target, given the noisy initial condition:

$$\mathcal{L}_{\text{PLM}} = \mathbb{E}_{t, p_1(\mathbf{x}_1), p_0(\mathbf{x}_0)} = [||u_t(\mathbf{x}_t|\mathbf{x}_1, \mathbf{x}_0) - v_\theta(\mathbf{x}_t, t)||^2] \tag{5}$$

After training, we can perform sampling from the PLM latent space by solving the ODE with a specified number of steps.

Additionally, we train an extra decoder to map the sampled PLM embeddings back to discrete sequences (details are provided in the Appendix).

### 3.3 FLOW MATCHING WITH COMBINED STRUCTURE AND PLM EMBEDDINGS

By integrating these two modalities, we obtain the final multi-modal CFM objective, which is a weighted sum of the individual flow matching objectives:

$$\mathcal{L}_{\text{CFM}} = \mathbb{E}_t(\lambda^{\text{SE(3)}}\mathcal{L}_{\text{SE(3)}} + \lambda^{\text{PLM}}\mathcal{L}_{\text{PLM}}) \tag{6}$$

To reduce chain breaks and steric clashes in the generated structures, we additionally incorporate a pairwise atomic distance-based local loss $\mathcal{L}_{\text{2D}}$ as an auxiliary loss term. Furthermore, to ensure semantic consistency between the predicted PLM embeddings and the original PLM embeddings, we also include a cosine similarity loss $\mathcal{L}_{\text{cos}}$. Additionally, we incorporate a cross-entropy loss $\mathcal{L}_{\text{Dec}}$ from the jointly trained decoder. Therefore, the total loss is the weighted sum of all loss terms:

$$\mathcal{L} = \mathcal{L}_{\text{CFM}} + \lambda^{\text{2D}}\mathcal{L}_{\text{2D}} + \lambda^{\text{cos}}\mathcal{L}_{\text{cos}} + \lambda^{\text{Dec}}\mathcal{L}_{\text{Dec}} \tag{7}$$

Additional details regarding model architecture, training configurations, and inference procedures are provided in the Appendix.

## 4 EXPERIMENTS

In this section, we evaluate the capability of IgFlow-LM in both unconditional design of antibody variable domains and conditional design of CDR loops. IgFlow-LM is trained on antibody structures from SabDab (Dunbar et al., 2014). To prevent data leakage, we partition the dataset based on the germline families of the antibodies, ensuring that no identical heavy-light chain germline family combinations appear in both the training and test sets.

### 4.1 UNCONDITIONAL DESIGN OF ANTIBODY VARIABLE DOMAINS

In this section, we evaluate the capability of IgFlow-LM in the unconditional generation of paired heavy and light chain variable domains. We adopt the same combinations of heavy and light chain lengths as used in IgDiff (Cutting et al., 2025). For each length combination, we sample 8 structures, resulting in a total of 2000 generated structures.

We adopted the novelty and diversity metrics proposed in IgDiff to evaluate the feasibility of IgFlow-LM for unconditional antibody design. Novelty is determined by calculating the RMSD of the HCDR3 loop relative to its nearest match, serving to quantify the degree of distinctness between the generated structures and known structures within the training set. Specifically, we first employed ANARCI (Dunbar & Deane, 2016) to annotate the sequences generated by IgFlow-LM, thereby

identifying the boundaries of all CDR regions. Subsequently, the nearest match was identified using the TM-score (Zhang & Skolnick, 2004) among entries with identical HCDR3 lengths. Diversity measures the model's capacity to yield multiple conformationally distinct solutions under identical design conditions. We calculated the pairwise RMSD between all CDR loops of the same type and length, following the same RMSD calculation methodology utilized for the novelty assessment.

Figure 2 (left) shows the HCDR3 RMSD distribution of antibodies generated by IgFlow-LM (1.7 Å) and the HCDR3 RMSD distribution of antibodies in the test set with the same HCDR3 loop lengths (0.91 Å). We additionally display the HCDR3 RMSD distribution of IgFlow-LM-generated antibodies relative to the test set (1.05 Å). The distributions indicate that IgFlow-LM is capable of generating novel HCDR3 loops based on the shape of their distribution

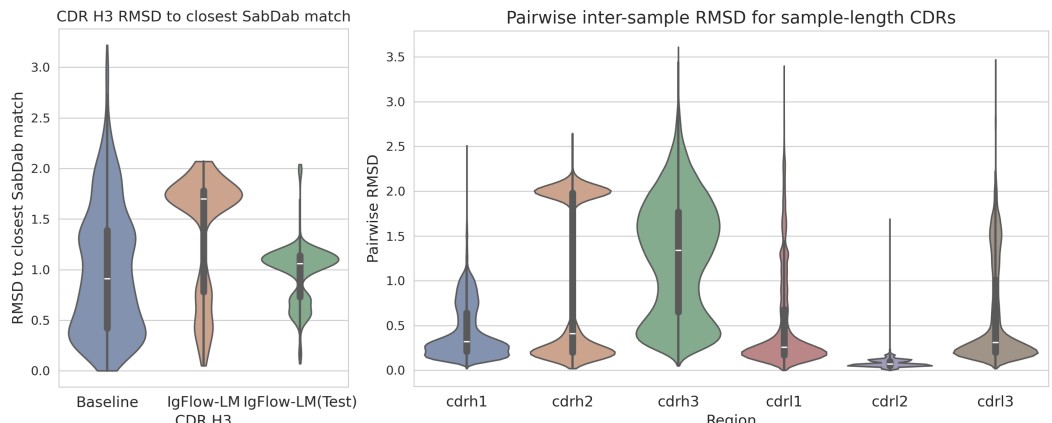

Figure 2: Novelty and diversity metrics computed from 2000 sampled structures. Left:Comparison of novelty between HCDR3 structures generated by IgFlow-LM and randomly selected samples from the training set. For each generated structure, the displayed RMSD corresponds to the structural deviation from its closest match in the training set (SAbDab) as determined by the TM-score, considering only structures with matching HCDR3 loop lengths. Right:Pairwise RMSD values among CDR loops of identical length generated by IgFlow-LM, illustrating the conformational diversity within each CDR type.

Figure 2 (right) shows the distribution of CDR regions across antibody structures generated by IgFlow-LM. We observe that HCDR3 exhibits the highest diversity, which is consistent with findings reported in previous studies (Nagaraj et al., 2024; Cutting et al., 2025; D'Angelo et al., 2018). Additionally, we note that LCDR2 demonstrates the lowest diversity, likely due to its relatively lower mutability compared to other CDR regions (Sankar et al., 2020; Teixeira et al., 2022). In summary, these results demonstrate that IgFlow-LM is capable of designing diverse CDR conformations across different loop regions while adhering to the general structural properties characteristic of each CDR type.

We next analyze the design capabilities of IgFlow-LM. For this evaluation, we select IgDiff (Cutting et al., 2025) and IgFlow (Nagaraj et al., 2024) as baseline models. We first assess the self-consistency of the model. To mitigate potential biases introduced by any single folding algorithm, we employ three independent structure prediction tools—ABodyBuilder2 (ABB2) (Abanades et al., 2023), Ig-Fold (Ruffolo et al., 2023), and ESMFold (Lin et al., 2023)—to fold the generated sequences (see Appendix for details on the folding models). For each folding model, we compute the root-mean-square deviation (RMSD) between the predicted structure and the original sampled structure, which we refer to as the self-consistency RMSD (scRMSD).It should be noted that IgDiff is not capable of simultaneous sequence–structure co-design; therefore, we utilize AbMPNN (Dreyer et al., 2023) to design heavy and light chain sequences for IgDiff, followed by structural folding of the designed sequences.

The results are presented in Table 1. As shown, IgFlow-LM achieves substantially higher success rates across the six CDR regions—0.82, 0.88, and 0.76—compared to IgFlow (0.63, 0.68, and 0.61), although it remains slightly below IgDiff (0.92, 0.96, and 0.80). We attribute IgDiff's superior

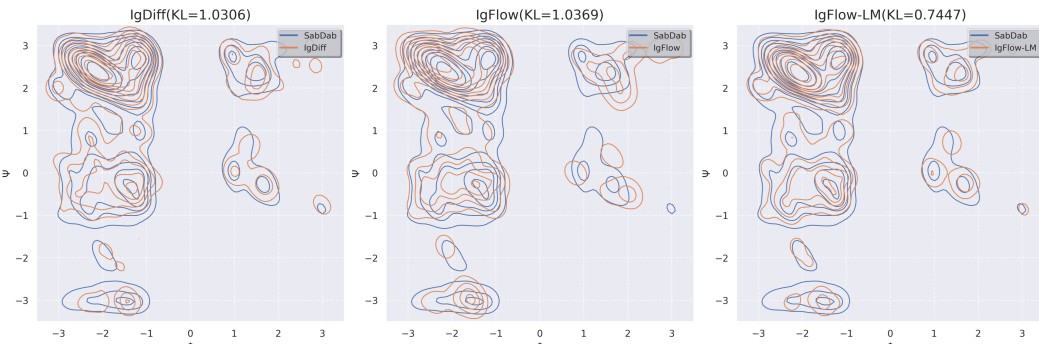

Figure 3: Ramachandran plots (Goodman et al., 1970) comparing the backbone dihedral angle distributions of antibody variable domains generated by IgFlow and IgFlow-LM against those in the SAbDab training set. The KL divergence value in each subplot's title indicates the Kullback–Leibler divergence between the generated structures and the SAbDab reference distribution.

performance to the fact that it was trained on structures folded using ABB2, leading to a closer alignment between its training distribution and that of antibody-specific folding models. Consequently, IgDiff exhibits a noticeable drop in scRMSD when evaluated with the general-purpose protein folding model ESMFold. Nevertheless, IgFlow-LM demonstrates strong design capabilities overall.

Table 1: Success rates of 2000 unconditional samples generated by IgFlow-LM and IgFlow, evaluated using two scRMSD metrics across two different folding models: ABB2 and ESMFold. Joint CDR scRMSD is defined as the fraction of samples where the RMSD of all six CDR loops between the refolded and sampled structures is below 2Å. CDR H3 scRMSD measures the fraction of samples where the RMSD of the HCDR3 loop between the refolded and sampled structures is below 2Å.

| Folding Model | | IgDiff | IgFlow | IgFlow-LM |
|---|---|---|---|---|
| ABB2 | Joint CDR scRMSD | 0.92 | 0.63 | 0.82 |
| | CDR H3 scRMSD | 0.93 | 0.69 | 0.84 |
| IgFold | Joint CDR scRMSD | 0.96 | 0.68 | 0.88 |
| | CDR H3 scRMSD | 0.97 | 0.73 | 0.89 |
| ESMFold | Joint CDR scRMSD | 0.77 | 0.61 | 0.77 |
| | CDR H3 scRMSD | 0.86 | 0.69 | 0.81 |

We then further visualize the backbone dihedral angle distributions of the structures generated by IgFlow-LM and IgFlow. The results are shown in Figure 3. It can be observed that IgFlow-LM generates antibody structures that closely adhere to the allowed distributions of backbone dihedral angles. Moreover, IgFlow-LM covers a broader range of low-density regions compared to IgFlow. Additionally, IgFlow-LM achieves a Kullback–Leibler (KL) divergence of 0.4309, which is significantly lower than that of IgDiff (1.0306) and IgFlow (1.0369). This indicates that the antibody conformations generated by IgFlow-LM—benefiting from its joint modeling with a protein language model—are more consistent with naturally observed antibody structures. We further report the deviations of the designed structures from ideal bond lengths and bond angles for both IgFlow-LM and IgFlow in Figure 6.

To further investigate the capability of IgFlow-LM in designing stable antibody structures, we employed PyRosetta (Chaudhury et al., 2010) to compute the interfacial binding free energy ($\Delta G$) between the heavy and light chains of antibodies designed by IgDiff, IgFlow, and IgFlow-LM, serving as a metric for structural stability. Notably, in addition to side-chain repacking performed via PyRosetta, all designed antibody structures underwent a relaxation procedure to relieve steric clashes and refine local geometry. To mitigate stochastic variability, we computed $\Delta G$ ten times for each antibody and reported the average value as its final $\Delta G$. As shown in Table 4, antibodies designed by IgFlow-LM exhibit the lowest average $\Delta G$, indicating stronger interfacial stability. Moreover,

IgFlow-LM also achieves the smallest RMSD between pre- and post-relaxation structures, suggesting that the sequence–structure pairs generated through its joint modeling with a protein language model (PLM) are structurally more consistent and physically plausible.

Subsequently, to demonstrate that the diversity introduced by IgFlow-LM does not significantly compromise the humaneness of the designed antibodies, we employed AbNatiV2 (Ramon et al., 2025) to evaluate the humaneness of sequences generated by both IgFlow and IgFlow-LM. We also incorporated IgDiff as a comparative benchmark representing an alternative antibody design pipeline. As presented in Table 5 and 6, although the composite humaneness score for VH-VL pairing of IgFlow-LM ranks second only to that of IgFlow, IgFlow-LM exhibits superior humaneness scores across the majority of CDR regions. Notably, it significantly outperforms other models in the light chain CDRs and the heavy chain H1/H2 regions. These findings indicate that, through synergistic modeling with the PLM, IgFlow-LM generates antibody sequences that more closely resemble natural human antibodies, thereby offering the potential to mitigate immunogenicity risks.

The sequence diversity of CDR loops, particularly the CDR H3 region, enables antibodies to recognize and respond to a wide variety of pathogens and antigens, thereby enhancing the breadth and depth of the immune response. This is critical for host immune defense, as it provides more comprehensive protection against pathogen escape and mutation (Schmitz et al., 2022; Dong et al., 2019).Finally, to demonstrate that integrating a PLM enhances IgFlow-LM's sequence design capability in unconditional design tasks, we further compare the diversity of sequences generated by IgFlow-LM and IgFlow. As previously stated, our analysis focuses on the HCDR3 region. The results are shown in Figure 4, which illustrates that IgFlow exhibits lower sequence diversity compared to IgFlow-LM. We also computed the positional entropy of the CDR H3 sequences designed by both models, yielding values of 2.1391 for IgFlow-LM and 1.2028 for IgFlow. Taken together, these results indicate that, with the assistance of a PLM, IgFlow-LM is capable of generating CDR loop sequences with significantly higher diversity. Furthermore, we performed t-SNE (Maaten & Hinton, 2008) dimensionality reduction to visualize the PLM latent embeddings of antibody sequences generated by IgFlow-LM in comparison with those of real antibody sequences, as shown in Figure 5. We also computed the canonical correlation analysis (CCA) score between the PLM latent representations of the generated and real sequences, yielding a value of 0.9213±0.0211 , along with a distance matrix correlation coefficient of 0.9081±0.0311 . These results indicate that IgFlow-LM effectively captures the underlying distribution of PLM latent embeddings observed in natural antibody sequences.

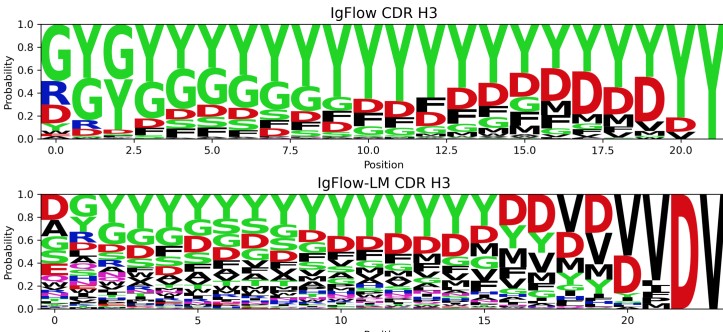

Figure 4: Sequence logo of the HCDR3 region generated by IgFlow-LM and IgFlow in the unconditional design task.

## 4.2 CONDITIONAL DESIGN OF CDR LOOPS

In antibody design engineering, the design of CDR loops represents a critical challenge, enabling the generation of diverse target antibody libraries in drug discovery (Mahajan et al., 2022). In this study, we consider two CDR loop design tasks: full design of all six CDRs conditioned on the remaining parts of the variable domain, and HCDR3 design conditioned on the other CDRs and the rest of the variable domain. It is important to note that the lengths of all CDR regions are kept fixed throughout.

Moreover, only the CDR regions are subject to design, while the remaining portions of the variable domain are held constant.

First, we evaluated the CDR-loop design capabilities of RFantibody (Bennett et al., 2025), DiffAb (Luo et al., 2022), IgFlow, and IgFlow-LM using scRMSD. It should be noted that none of the aforementioned models incorporate antigen information when modeling CDR regions. We selected 100 antibody structures from the test set constructed above, which collectively span the common lengths of antibody variable regions and include non-canonical CDR loop lengths. For each reference structure, we generated 20 designed structures. Subsequently, we folded all designed sequences using a folding model and computed the RMSD between each predicted structure and the corresponding original sampled structure.

The results for the design of the full CDR regions and the CDR H3 region are presented in Tables 3 and 7, respectively. It can be observed that IgFlow-LM exhibits consistent performance regardless of the folding model employed. In contrast, the other baseline models show significant performance discrepancies between the general protein folding model, ESMFold, and the two antibody-specific folding models. This suggests that IgFlow-LM, through its synergistic modeling with the PLM, demonstrates greater stability compared to the baseline models.

Table 2: Comparison of self-consistency RMSD (scRMSD) success rates across the six CDR regions for RFantibody, DiffAb, IgFlow, and IgFlow-LM in conditional CDR design tasks. Higher scRMSD success rates indicate better designability in the given region. "All" refers to the proportion of samples where all six CDRs loops are independently considered designable.

| Folding Model | | All | HCDR1 | HCDR2 | HCDR3 | LCDR1 | LCDR2 | LCDR3 |
|---|---|---|---|---|---|---|---|---|
| ABB2 | RFantibody | 0.84 | 0.99 | 0.99 | 0.86 | 0.97 | 1.00 | 0.99 |
| | DiffAb | 0.90 | 0.99 | 0.99 | 0.90 | 0.99 | 1.00 | 1.00 |
| | IgFlow | 0.88 | 0.99 | 0.99 | 0.89 | 0.99 | 1.00 | 0.99 |
| | IgFlow-LM | 0.88 | 1.00 | 0.99 | 0.88 | 0.99 | 1.00 | 1.00 |
| IgFold | RFantibody | 0.89 | 1.00 | 0.99 | 0.90 | 0.99 | 1.00 | 1.00 |
| | DiffAb | 0.92 | 0.99 | 0.99 | 0.93 | 0.99 | 1.00 | 1.00 |
| | IgFlow | 0.89 | 0.99 | 0.99 | 0.90 | 1.00 | 1.00 | 0.99 |
| | IgFlow-LM | 0.89 | 1.00 | 0.99 | 0.90 | 1.00 | 1.00 | 1.00 |
| ESMFold | RFantibody | 0.92 | 1.00 | 0.99 | 0.95 | 0.97 | 1.00 | 0.99 |
| | DiffAb | 0.86 | 0.99 | 0.99 | 0.89 | 0.96 | 1.00 | 0.99 |
| | IgFlow | 0.83 | 0.99 | 0.99 | 0.86 | 0.96 | 1.00 | 0.99 |
| | IgFlow-LM | 0.86 | 1.00 | 0.99 | 0.87 | 0.97 | 1.00 | 0.99 |

To further validate the hypothesis regarding the enhanced stability of IgFlow-LM, we employed Cochran's Q test to assess the overall variance of each generative model across different folding backends, followed by pairwise McNemar's tests with Holm correction. We evaluated the performance metrics for all models in the HCDR3 design task. The results are summarized in Table 8. For all four generative models, Cochran's Q test indicated no overall significant difference among the folding backends (all p-values = 1.0). regarding the IgFlow-LM model, only the comparison between ESMFold and IgFold revealed a significant difference after correction (p = 0.0044), whereas ABB2 showed no significant difference with either. The success rates consistently remained within a narrow range (84.7%–87.5%, with overlapping 95% confidence intervals), demonstrating the high robustness of IgFlow-LM across different folding models. In contrast, the other generative models (IgFlow, DiffAb, and RFantibody) exhibited multiple significant pairwise differences and greater variability across folding models, suggesting lower consistency in structural fidelity.

Subsequently, we employed PyRosetta to calculate the heavy-light chain interface $\Delta$G and the Relax RMSD for the CDR regions redesigned by RFantibody, DiffAb, IgFlow, and IgFlow-LM. The evaluation protocol included side-chain repacking (with the exception of DiffAb) followed by structural relaxation. The results are summarized in Tables 9 and 10. Notably, IgFlow-LM exhibited the lowest interface $\Delta$G and the minimal structural deviation upon relaxation (Relax RMSD) among all evaluated models.

To further investigate the impact of the PLM on the diversity of CDR regions designed by IgFlow-LM, we compared the amino acid diversity of sequences generated during conditional CDR loop design against those from DiffAb and IgFlow. The HCDR3 diversity results are presented in Figure 5. It can be observed that IgFlow-LM exhibits superior overall diversity compared to DiffAb and IgFlow, and more closely resembles the distribution of native antibodies. Furthermore, we calculated the positional information entropy for the HCDR3 sequences designed by DiffAb, IgFlow, and IgFlow-LM, yielding values of 2.1435, 1.4333, and 1.9758, respectively. Collectively, these results indicate that, aided by the PLM, IgFlow-LM is capable of designing CDR loop sequences with high diversity.

Additionally, we used AbNatiV2 to compute humanization scores for the CDR regions redesigned by RFantibody, DiffAb, IgFlow, and IgFlow-LM. The results are shown in Tables 11 to 13. IgFlow-LM again demonstrates the highest overall humanization score.

### 4.3 ABLATION STUDY

To further investigate the impact of integrating a PLM into the generative framework on antibody design performance, we conducted an ablation study in which the PLM probability flow is removed. Specifically, during inference, we perform ODE evolution only on the SE(3) backbone frames, while omitting the ODE integration over the PLM latent flow. In both cases, the same decoder is used to map the final generated PLM embeddings to amino acid sequences. The variant without PLM latent flow evolution is denoted as IgFlow-LM (w/o PLM latent Flow).

As shown in Table 3, disabled the evolution of the ODE of the PLM probability flow leads to a noticeable degradation in design performance. This indicates that when relying solely on structural signals, the model tends to generate conformations that are structurally plausible but may yield sequences that are less natural or evolutionarily inconsistent. The results suggest that the PLM probability flow plays an important role in preserving sequence plausibility and diversity during the generation process.

Table 3: Success rates of 2000 unconditional samples generated by IgFlow-LM and IgFlow-LM (w/o PLM latent Flow), evaluated using two scRMSD metrics across two different folding models: ABB2 and ESMFold. Joint CDR scRMSD is defined as the fraction of samples where the RMSD of all six CDR loops between the refolded and sampled structures is below 2Å. CDR H3 scRMSD measures the fraction of samples where the RMSD of the HCDR3 loop between the refolded and sampled structures is below 2Å.

| Folding Model | | IgFlow-LM (w/o PLM latent Flow) | IgFlow-LM |
|---|---|---|---|
| ABB2 | Joint CDR scRMSD | 0.59 | 0.64 |
| | CDR H3 scRMSD | 0.60 | 0.72 |
| IgFold | Joint CDR scRMSD | 0.61 | 0.69 |
| | CDR H3 scRMSD | 0.61 | 0.75 |
| ESMFold | Joint CDR scRMSD | 0.55 | 0.61 |
| | CDR H3 scRMSD | 0.58 | 0.71 |

## 5 CONCLUSION

In this work, we integrate a probabilistic flow from a protein language model with an SE(3) flow-matching architecture to design antibody variable domains under both conditional and unconditional settings. We find that IgFlow-LM is capable of generating realistic antibodies consistent with its training distribution and produces structures that better conform to physical constraints. Under conditional design, IgFlow-LM outperforms baseline models in overall structural quality and also demonstrates superior sequence diversity compared to all other baselines. Future work will involve training IgFlow-LM on a larger number of experimentally determined and synthetic antibody structures, exploring improved strategies for integrating protein language models with structure design, and investigating antigen-aware co-design of antibody sequences and structures.

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

# A   MODEL ARCHITECTURE DETAILS

## A.1   REPRESENTATIONS

**SE(3)-Flow Matching.** In three-dimensional structural modeling, the spatial conformation of a protein is determined by the rotations and translations of its rigid groups. These transformations are commonly described using two standard mathematical groups: SO(3) and SE(3).

SO(3) (the three-dimensional special orthogonal group) represents the set of all three-dimensional rotations and consists of all 3×3 orthogonal matrices with determinant 1. It encompasses only rotational transformations, excluding translations, and preserves both distances and angles. In protein modeling, SO(3) is typically used to describe the rotation of local residue coordinate frames. SE(3) (the three-dimensional special Euclidean group) represents the set of all three-dimensional rigid-body transformations, formed by combining rotations from SO(3) with translations in $\mathbb{R}^3$. It includes both rotational and translational components and is used to describe the full rigid-body pose of atomic groups or residues in space. Modern structure prediction models, such as AlphaFold and RoseTTAFold, operate directly in SE(3) space.

In IgFlow-LM, we utilize the SE(3)-equivariant flow matching framework introduced in FrameFlow Yim et al. (2023a) to model the backbone structure of antibodies. For backbone parameterization, we directly adopt the framework used in AlphaFold2. According to the description in AlphaFold2, each residue in an antibody can be represented by a rigid-body frame consisting of a rotation matrix $r_i \in$ SO(3) and a translation vector $x_i \in \mathbb{R}^3$, which together form a group element Hu et al. (2024). The backbone of an antibody with $N$ residues can then be represented as the sequence of these frames $\{T^{(1)}, ..., T^{(N)}\} \in \text{SE}(3)^N$. It is important to note that, following the convention in FrameFlow, we represent protein structures within the zero-center-of-mass (CoM) subspace of $\mathbb{R}^{N \times 3}$ by subtracting the CoM from all data points. This step is crucial for ensuring that the distribution of generated frames remains SE(3)-equivariant.

Based on the work by Yim et al. (2023b), we can independently model SO(3) and $\mathbb{R}^3$ during both training and sampling, meaning that these two components can be modeled separately. Following the setup in FrameFlow, the prior distribution for SO(3) is chosen as a uniform distribution over SO(3), while the prior distribution for $\mathbb{R}^3$ is selected as an isotropic Gaussian distribution $\mathcal{N}(0, I_3)$. Therefore, the prior distribution for a single frame can be set as $p_0(T_0^{(i)}) = \mathcal{U}(\text{SO}(3)) \otimes \mathcal{N}(0, I_3)$. The conditional flow $T_t = \psi_t(T_0|T_1)$ is then defined as the geodesic between $T_0$ and $T_1$:

$$T_t = \exp_{T_0}(\log_{T_0}(T_1)) \tag{8}$$

where $\exp_T$ denotes the exponential map at point $T$, and $\log_T$ represents the logarithmic map at point $T$. By applying the logarithmic map to obtain the tangent vector from $T_0$ to $T_1$, scaling it by the factor $t$, and then transforming it back to the manifold using the exponential map, the resulting point lies on the geodesic. According to the work by Yim et al. (2023b), Equation (1) can be simplified into the following forms:

$$\text{Translations}(\mathbb{R}^3) : x_t = (1 - t)x_0 + tx_1 \tag{9}$$

$$\text{Rotations}(\text{SO}(3)) : \exp_{r_0}(t\log_{r_0}(r_1)) \tag{10}$$

The exponential map $\exp_{r_0}$ can be directly computed using the Rodrigues formula, while the logarithmic map $\log_{x_0}$ is calculated based on the results from Yim et al. (2023b). Therefore, the overall objective can be written as:

$$\mathcal{L}_{\text{SE}(3)} = \mathbb{E}_{t, p_0(\mathbf{T}_0), p_1(\mathbf{T}_1)}[\sum_{n=1}^{N}\{||v_x^{(n)}(\mathbf{T}_t, t) - x_t^{(n)}||_{\mathbb{R}^{\mu}}^2 + \\ ||v_r^{(n)}(\mathbf{T}_t, t) - r_t^{(n)}||_{\text{SO}(3)}^2\}] \tag{11}$$

where $N$ represents the number of residues in the antibody. This objective approximates the CFM target mentioned in our manuscript. The vector fields $v_x^{(n)}(\mathbf{T}_t, t)$ and $v_r^{(n)}(\mathbf{T}_t, t)$ can be fitted using SE(3)-equivariant neural networks. And we calculate the derivatives of $x_t^{(n)}$ and $r_t^{(n)}$ with respect to time $t$:

$$x_t^{(n)} = \frac{x_1^{(n)} - x_t^{(n)}}{1 - t}, r_t^{(n)} = \frac{\log_{r_t^{(n)}}(r_1^{(n)})}{1 - t} \tag{12}$$

We define $\{(\hat{x}_1^{(n)}, \hat{r}_1^{(n)})\}_{n=1}^N$ as the interpolated frames $\mathbf{T}_t$ at time t , which are predictions based on the original frame $\mathbf{T}_1$. According to the work by Yim et al. (2023b), we reparameterize the objective as:

$$\mathcal{L}_{\text{SE(3)}} = \mathbb{E}_{t,p_0(\mathbf{T}_0),p_1(\mathbf{T}_1)}\Big[\frac{1}{(1-t)^2}\sum_{n=1}^N\{||\hat{x}_1^{(n)}(\mathbf{T}_t,t)-$$

$$x_1^{(n)}||_{\mathbb{R}^{}}^2 + ||\log_{r_t^{(n)}}(\hat{r}_1^{(n)}(\mathbf{T}_t,t)) - \log_{r_t^{(n)}}(r_1^{(n)})||_{\text{SO(3)}}^2\}\Big] \tag{13}$$

**PLM-based Probability Flows Matching for PLM Embeddings.** PLMs convert discrete amino acid sequences into continuous numerical representations (Ding et al., 2019), allowing us to directly define a vector field between the prior distribution $p_0(\mathbf{x}_0) = p_{\text{noise}}(\mathbf{x}_0)$ and the target distribution $p_1(\mathbf{x}_1) = p(\mathbf{x})$ based on the FM principle:

$$\frac{d\mathbf{x}_t}{dt} = v_\theta(\mathbf{x}_t, t) \tag{14}$$

Here, we choose an isotropic Gaussian distribution $\mathcal{N}(0, I_D)$ as the prior distribution for PLM embeddings, where $D$ denotes the dimensionality of the PLM embeddings. The corresponding flow can be defined as a conditional Gaussian distribution (Song et al., 2020b):

$$p_t(\mathbf{x}_t) = \mathbb{E}_{\mathbf{x}_t \sim \mathcal{N}(\mu_t, \sigma_t)}p(\mathbf{x}_1|\mathbf{x}_0) = \mathcal{N}(\mathbf{x}_1|\mu_t(\mathbf{x}_t), \sigma_t(\mathbf{x}_1)) \tag{15}$$

This form is widely used in existing generative models (Ho et al., 2020). Similarly, we can define the following flow matching objective for the vector field $u_t(\mathbf{x}_t)$:

$$\mathcal{L}_{\text{FM}} = \mathbb{E}_{t,p_t(\mathbf{x}_t)}||v_\theta(\mathbf{x}_t, t) - u_t(\mathbf{x}_t)||^2 \tag{16}$$

Empirically, we construct a conditional vector field $u_t(\mathbf{x}_t|\mathbf{x}_1, \mathbf{x}_0)$:

$$u_t(\mathbf{x}_t|\mathbf{x}_1, \mathbf{x}_0) = (1-t)\mathbf{x}_0 + t\mathbf{x}_1 \tag{17}$$

This form intuitively constructs a geodesic (i.e., a straight line segment) between two points in Euclidean space, which aligns with some existing flow matching losses used for image generation (Chen & Lipman, 2023). At this point, we reparameterize the original flow matching objective as follows:

$$\mathcal{L}_{\text{PLM}} = \mathbb{E}_{t,p_1(\mathbf{x})_1,p_0(\mathbf{x})_0}||u_t(\mathbf{x}_t|\mathbf{x}_1, \mathbf{x}_0) - v_\theta(\mathbf{x}_t, t)||^2 \tag{18}$$

**Decoder for Mapping PLM Embeddings Back to Discrete Sequence Space.** After generating new PLM embeddings using IgFlow-LM, we require a decoder to map these embeddings back to the discrete sequence space, thereby obtaining complete new antibody sequence-structure pairs. The decoder maps the generated PLM embeddings $\hat{\mathbf{x}}_1 = [\hat{\mathbf{x}}_1^{(1)}, ..., \hat{\mathbf{x}}_1^{(L)}]$ to a discrete sequence:

$$\hat{y}^{(i)} = \mathcal{L}_{\mathbb{D}} = \text{Softmax}(\mathbf{W}_0 \cdot f_\phi(\hat{\mathbf{x}}_1^{(i)}) + \mathbf{b}_0) \tag{19}$$

where $f_\phi$ represents a multi-layer perceptron (MLP), $\mathbf{W}_0$ and $\mathbf{b}_0$ are the projection matrix and bias term of the output layer, respectively, and $\hat{y}^{(i)}$ denotes the predicted amino acid probability distribution at position $i$. We optimize this decoder using cross-entropy loss:

$$\mathcal{L}_{\mathbb{D}} = -\frac{1}{L}\sum_{l=1}^L \log\hat{p}_\phi(a_l|\hat{\mathbf{x}}_1^{(l)}) \tag{20}$$

where $\hat{p}_\phi(a_l|\hat{\mathbf{x}}_1^{(l)})$ represents the decoder's predicted probability distribution for the $l$-th amino acid, and $a_l$ denotes the true value of the $l$-th amino acid.

## A.2 MODEL ARCHITECTURE

IgFlow-LM is a modified version of the FrameFlow architecture (Yim et al., 2023a), adapted for sequence-structure co-design of antibodies. We set the hidden dimension of the IPA to 16 and increased the number of IPA modules from 6 to 10. Additionally, we increased the number of transformer layers per block from 2 to 4, thereby enhancing the overall depth of the network to better accommodate the multimodal task of jointly generating PLM embeddings and antibody structures.

Dropout was applied to each transformer layer with a dropout rate of 0.1, randomly setting 0.1 of the neurons to zero during training to mitigate overfitting.

**Encoder.** The model's encoder takes as input the three-dimensional structure of the target protein and outputs node embeddings and edge (residue-pair) embeddings. For node (residue) embeddings, we use a combination of the following features:

- Interpolated PLM embeddings.
- Positional encoding of residuals.
- Chain index encoding.
- Timestep embedding.

In the case of conditional CDR region design for antibodies, additional features derived from non-CDR regions are incorporated:

- Residue type embeddings.
- Atomic coordinates, including both backbone and side-chain atoms.
- The dihedral angles of the backbone, encoded using sine-based positional embeddings.
- Side-chain torsion angles, similarly encoded with sine-based embeddings.

Each feature is first processed through separate MLPs. The resulting representations are concatenated and further transformed via an additional MLP to produce the final node embeddings.

For edge (residue-pair) embeddings, we employ a mixture of the following features:

- Cross-concatenation of the two node embeddings.
- Relative positional encoding between residues.
- Positional bins of the interpolated translation vector between residue pairs.

Additionally, for conditional CDR design in antibodies, the following non-CDR-derived pairwise features are included:

- Residue type pair embeddings, represented by a learnable $20 \times 20$ dimensional embedding matrix.
- Relative sequence position, modeled via learnable embeddings of the sequence distance between residue pairs.
- Euclidean distance between the two residues.
- Relative orientation between residues, characterized by computing inter-residue backbone dihedral angles and encoded using sine-based embeddings.

Similarly, these edge features are individually encoded through dedicated MLPs, concatenated, and passed through another MLP to generate the final edge embeddings.

Similarly, these features are encoded using different MLPs and are concatenated features and transformed by another MLP to form the final edge embedding. The encoder is SE(3)-invariant, meaning that its output will be the same regardless of any global rigid transformation. Here we set the hidden dimension of residue embedding as 256, and the hidden dimension of pair embedding hidden dimension as 128.

**Protein Language Model.** For the choice of PLM, we select IgBert (Kenlay et al., 2024), which is specifically trained on the Observed Antibody Space (OAS) Olsen et al. (2022). IgBert can accept both heavy chain and light chain sequences as input and output the corresponding latent space embeddings. IgBert appends three special tokens—start, end, and chain separation—to the input sequences. The output is an embedding of size $\mathbb{R}^{(L+3) \times 1024}$, where L is the length of the antibody sequence, and 1024 is the dimensionality of IgBert. Since these special tokens may affect the alignment between sequences and structures, we remove the embeddings corresponding to these three special tokens before using them for model training.

We also employ self-conditioning based on distances and backbone dihedral angles. Specifically, during training, the predicted translation vectors and backbone coordinates, obtained with gradient computation disabled, are used as part of the model input. The translation vectors are then converted into discrete distance bin features, and backbone dihedral angles are computed from the backbone coordinates, both of which are incorporated as part of the initial encoding features.

Furthermore, we find that incorporating additional equivariant features—such as backbone dihedral angles and residue centrality—as well as invariant features—including relative sequence position and secondary structure information—is beneficial in the conditional CDR generation setting. In other words, during conditional CDR generation, information from the non-CDR regions within the entire variable domain is provided as additional conditioning input to the model.

## A.3 TRAINING DETAILS

**Dataset.** We trained IgFlow-LM on the Structural Antibody Database (SAbDab) (Dunbar et al., 2014). The antibody structures in this database are all experimentally resolved. We preprocessed each structure by retaining only those with a resolution below 4Å and trimming their variable regions. To avoid potential issues arising from unresolved residues in the structural data, we aligned sequences using the variable-region sequences provided by the SAbDab database as the reference. For residues lacking resolved coordinates, we assigned zero-valued coordinate placeholders and excluded these unresolved residues from loss computation.

To prevent data leakage, we partitioned the dataset based on the germline families of the antibodies, ensuring that no identical heavy-light chain germline family combinations appear in both the training and test sets. For antibodies lacking family annotations for either the heavy or light chain, we first attempted to assign their germline families using ANARCI. Only those sequences for which family assignment could not be determined were included in the training set. After this procedure, the final dataset consists of 726 test samples and 9,850 training samples. The germline family combinations reserved for the test set are as follows: IGHV1-IGKV10, IGHV14-IGKV6, IGHV1-IGKV12, IGHV5-IGKV3, IGHV14-IGKV14, IGHV4-IGLV2, IGHV3-IGLV6, IGHV2-IGKV5.

**Loss Function.** The flow matching-based loss comprises two components: the SE(3) flow-related loss and the PLM probability flow-related loss. The SE(3) flow-related loss includes terms for the predicted rotation matrices and translation vectors relative to their ground-truth values. The translation loss is computed as the L2 distance between the model-predicted and true translation vectors. The rotation loss is defined as the L2 loss on SO(3), measuring the discrepancy between predicted and ground-truth rotation matrices. Both losses are normalized by the number of valid residues during averaging. Their respective weights are set to 2.0 and 1.0. Additionally, we incorporate two structural auxiliary losses: a backbone atomic position loss and a local proximity loss based on pairwise interatomic distances, with weights set to 0.2 and 0.2, respectively. The PLM probability flow-related loss is defined as the L2 distance between the model-predicted and true PLM latent embeddings, again averaged over the number of valid residues. This term is assigned a weight of 0.02. To further regularize the PLM embedding predictions, we include a cosine similarity loss as an auxiliary objective, with a weight of 0.25.

The decoder is trained jointly with the flow matching model to strengthen the alignment between the generated embeddings and the sequence decoding process. The decoder is optimized using the cross-entropy loss, with a weight of 1.0.

We used the AdamW optimizer with a learning rate of , a batch size of 32, and weight decay set to 0.01. A cosine learning rate schedule was employed, with a warm-up phase for the first 500 optimization steps, followed by a gradual decay to over 30,000 optimization steps. Training was conducted on four A100 PCIe 40G GPUs, totaling 30,000 optimization steps. Additionally, during training, we randomly decided whether to use self-conditioning for model optimization with a probability of 0.5.

The code for IgFlow-LM can be accessed at https://anonymous.4open.science/r/IgFlow-LM-8848.

# B  APPENDIX

Table 4: Interfacial $\Delta$G between heavy and light chains and RMSD between pre- and post-relaxation structures for antibodies designed by IgDiff, IgFlow, and IgFlow-LM calculated using PyRosetta. We report the means and standard deviations for all metrics.

| Model | $\Delta$G | Relax RMSD |
|---|---|---|
| IgDiff | -57.2654±9.3825 | 1.04±0.2675 |
| IgFlow | -53.2978±9.8673 | 1.56±0.3358 |
| IgFlow-LM | -58.1383±7.9444 | 0.93±0.2158 |

Table 5: Humaneness scores of antibody sequences designed by IgFlow and IgFlow-LM evaluated using AbNatiV2. "Heavy-Light" denotes the humaneness score for the heavy-light chain pairing. All scores range from 0 to 1, where higher scores indicate a closer resemblance to natural human antibodies.

| Model | Heavy-Light | CDR H1 | CDR H2 | CDR H3 |
|---|---|---|---|---|
| IgDiff | 0.8486±0.0611 | 0.7755±0.1779 | 0.6329±0.2591 | 0.7588±0.1810 |
| IgFlow | 0.8664±0.0524 | 0.7179±0.2007 | 0.5635±0.1774 | 0.8295±0.1444 |
| IgFlow-LM | 0.8658±0.0802 | 0.8052±0.1940 | 0.7210±0.2467 | 0.7122±0.1699 |

Table 6: Humaneness scores of antibody sequences designed by IgFlow and IgFlow-LM evaluated using AbNatiV2. All scores range from 0 to 1, where higher scores indicate a closer resemblance to natural human antibodies.

| Model | CDR L1 | CDR L2 | CDR L3 |
|---|---|---|---|
| IgDiff | 0.6834±0.3131 | 0.7517±0.4110 | 0.6501±0.2782 |
| IgFlow | 0.5182±0.3487 | 0.7200±0.4305 | 0.7341±0.1941 |
| IgFlow-LM | 0.7452±0.3304 | 0.8192±0.2986 | 0.8088±0.2161 |

Table 7: Comparison of HCDR3 scRMSD success rates among RFantibody, DiffAb, IgFlow, and IgFlow-LM for the conditional CDR H3 design task. A higher scRMSD success rate indicates superior designability of the specified region.

| Folding Model | | HCDR3 |
|---|---|---|
| | RFantibody | 0.88 |
| ABB2 | DiffAb | 0.88 |
| | IgFlow | 0.89 |
| | IgFlow-LM | 0.87 |
| | RFantibody | 0.91 |
| IgFold | DiffAb | 0.90 |
| | IgFlow | 0.90 |
| | IgFlow-LM | 0.88 |
| | RFantibody | 0.94 |
| ESMFold | DiffAb | 0.87 |
| | IgFlow | 0.85 |
| | IgFlow-LM | 0.86 |

Table 8: Structural accuracy (scRMSD $\leq 2$Å) of designed HCDR3 regions across generative models and folding backends. Values are reported as success rate with 95% confidence interval. Statistical significance is evaluated using McNemar's test with Holm correction; $^*$ indicates $p_{\text{adj}} < 0.05$.

| Model | ESMFold | ABB2 | IgFold |
|---|---|---|---|
| IgFlow-LM | 0.847 (0.831–0.862) | 0.858 (0.841–0.872) | 0.875 (0.860–0.889)$^*$ |
| IgFlow | 0.853 (0.837–0.868)$^*$ | 0.900 (0.886–0.912)$^*$ | 0.902 (0.889–0.915)$^*$ |
| DiffAb | 0.877 (0.862–0.891)$^*$ | 0.884 (0.870–0.898)$^*$ | 0.902 (0.889–0.915)$^*$ |
| RFantibody | 0.940 (0.929–0.950)$^*$ | 0.878 (0.863–0.892)$^*$ | 0.915 (0.902–0.926)$^*$ |

Table 9: Heavy-light chain interface $\Delta$G and Relax RMSD calculated using PyRosetta for IgDiff, IgFlow, and IgFlow-LM in the 6-CDR redesign task. The RMSD reflects the structural deviation between the antibody structures before and after relaxation. We report the mean and standard deviation for all metrics.

| Model | $\Delta$G | Relax RMSD |
|---|---|---|
| RFantibody | -55.1967±8.8900 | 1.12±0.3438 |
| DiffAb | -56.0825±8.7548 | 1.10±0.3822 |
| IgFlow | -57.0922±8.5538 | 1.11±0.4084 |
| IgFlow-LM | -57.1340±8.2845 | 0.96±0.3221 |

Table 10: Heavy-light chain interface $\Delta$G and Relax RMSD calculated using PyRosetta for IgDiff, IgFlow, and IgFlow-LM in the HCDR3 redesign task. The RMSD reflects the structural deviation between the antibody structures before and after relaxation. We report the mean and standard deviation for all metrics.

| Model | $\Delta$G | Relax RMSD |
|---|---|---|
| RFantibody | -55.0949±9.1402 | 1.24±0.3460 |
| DiffAb | -54.4597±8.4309 | 1.01±0.4162 |
| IgFlow | -55.1652±8.8481 | 1.05±0.4147 |
| IgFlow-LM | -55.2103±8.8527 | 0.91±0.3110 |

Table 11: Humanness scores calculated using AbNatiV2 for antibody sequences generated by RFantibody, DiffAb, IgFlow, and IgFlow-LM in the 6-CDR redesign task. Scores range from 0 to 1, with higher values indicating a closer resemblance to native human antibodies.

| Model | CDR H1 | CDR H2 | CDR H3 |
|---|---|---|---|
| RFantibody | 0.4609±0.1877 | 0.4036±0.2419 | 0.3402±0.1748 |
| DiffAb | 0.7858±0.2048 | 0.5045±0.2240 | 0.6923±0.2025 |
| IgFlow | 0.8220±0.1769 | 0.7584±0.1716 | 0.7313±0.1802 |
| IgFlow-LM | 0.8501±0.1894 | 0.7896±0.2033 | 0.7024±0.1863 |

Table 12: Humanness scores calculated using AbNatiV2 for antibody sequences generated by RFantibody, DiffAb, IgFlow, and IgFlow-LM in the 6-CDR redesign task. Scores range from 0 to 1, with higher values indicating a closer resemblance to native human antibodies.

| Model | CDR L1 | CDR L2 | CDR L3 |
|---|---|---|---|
| RFantibody | 0.5331±0.2803 | 0.4641±0.3192 | 0.6012±0.2326 |
| DiffAb | 0.6396±0.3685 | 0.7389±0.2396 | 0.6636±0.2040 |
| IgFlow | 0.8580±0.3178 | 0.8586±0.1833 | 0.8451±0.1541 |
| IgFlow-LM | 0.8182±0.3492 | 0.9134±0.1771 | 0.8355±0.1651 |

Table 13: Humanness scores calculated using AbNatiV2 for antibody sequences generated by RFantibody, DiffAb, IgFlow, and IgFlow-LM in the CDR H3 redesign task. Scores range from 0 to 1, with higher values indicating a closer resemblance to native human antibodies.

| Model | CDR H3 |
|---|---|
| RFantibody | 0.3545±2003 |
| DiffAb | 0.6593±0.2074 |
| IgFlow | 0.7705±0.1680 |
| IgFlow-LM | 0.6937±0.2014 |

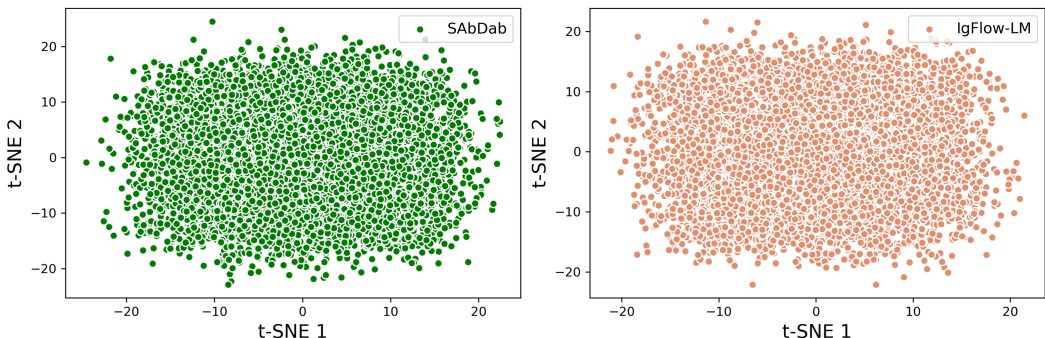

Figure 5: Visualization of IgBert embedding vectors generated by IgFlow-LM and reduced in dimension via t-SNE, alongside visualization of IgBert embedding vectors derived from SabDab antibody sequences.

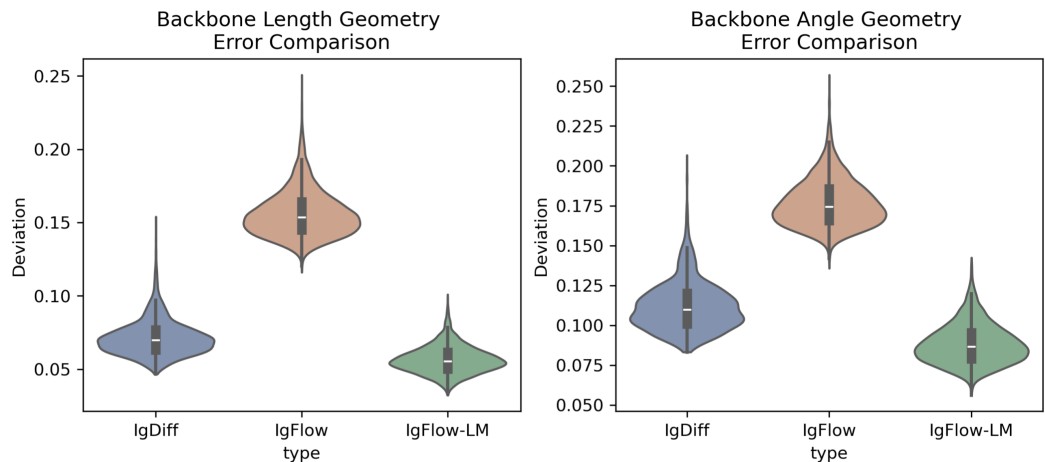

Figure 6: Deviations of bond lengths and dihedral angles from their ideal values in structures generated by IgDiff, IgFlow, and IgFlow-LM. The ideal bond length is 1.329 Å; the ideal cosine values for the dihedral angles CA-C-N and C-N-CA are -0.4415 and -0.5255, respectively.

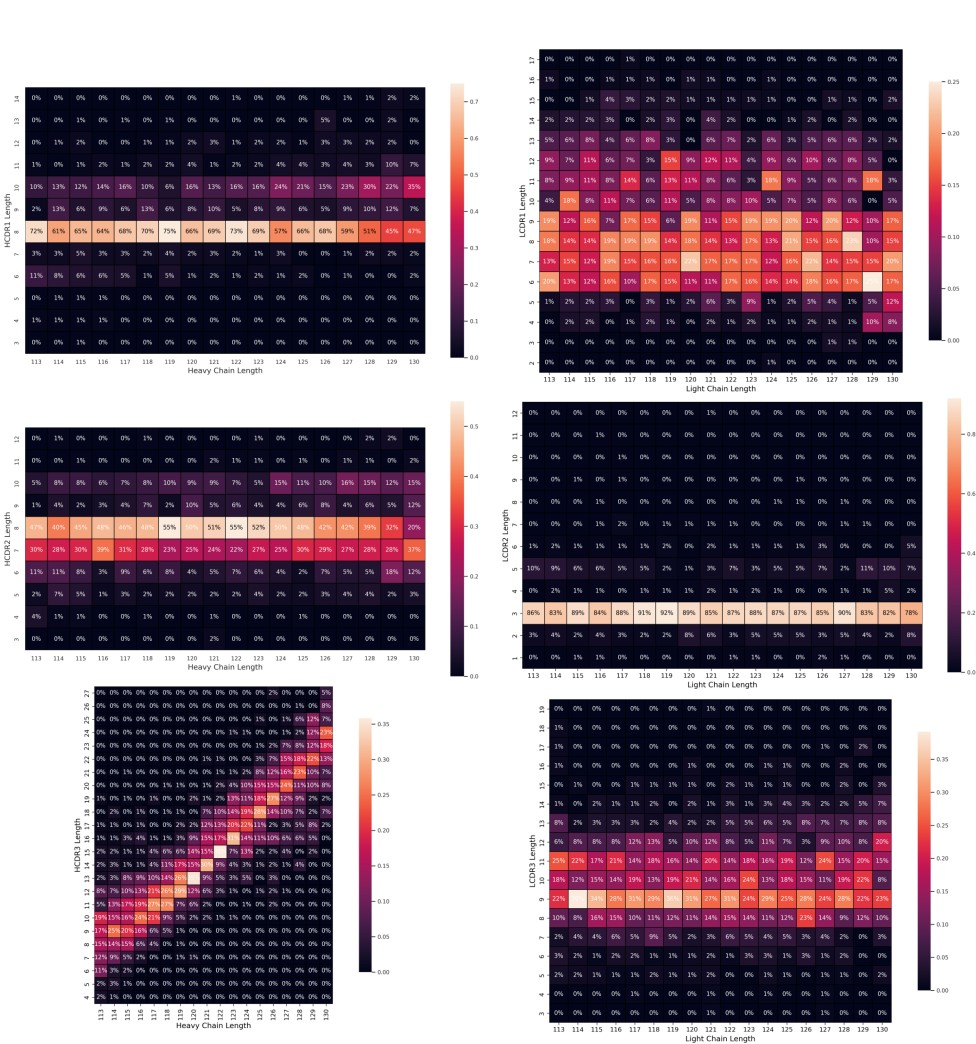

Figure 7: Distribution of CDR loop lengths across designs of specified heavy and light chain lengths. We note that HCDR length changes seem to directly correlate and possibly drive HCDR3 length, with smaller associations observed for the other chains.

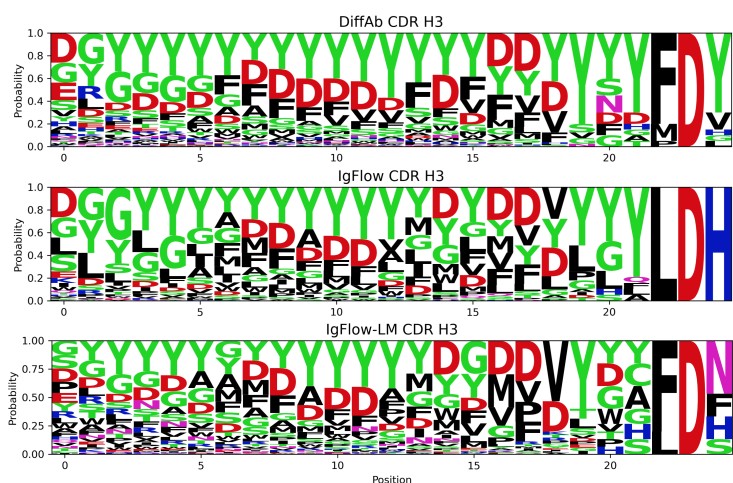

Figure 8: Sequence logo of the HCDR3 region generated by IgFlow-LM and IgFlow in the full CDR design task, depicting the frequency of amino acid occurrences at each position.

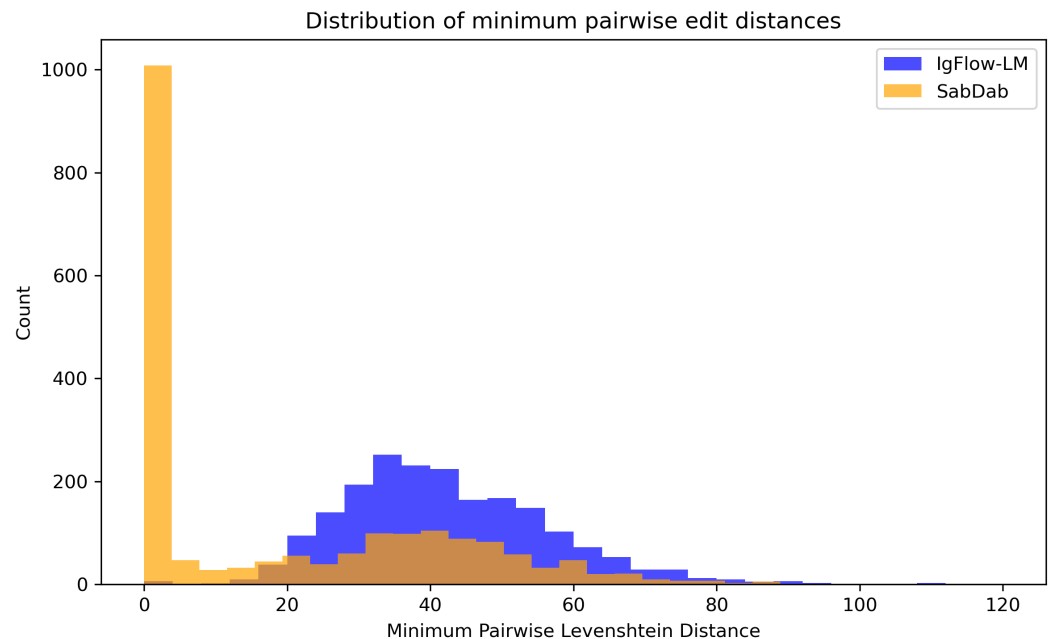

Figure 9: Histogram of the minimum pairwise Levenshtein distance between 2000 unconditioned IgFlow-LM generated structures (blue) and 2000 paired SabDab sequences (orange). SabDab contains experimentally determined antibody structures, many of which originate from similar immune responses or have undergone engineering modifications, leading to a significant degree of sequence redundancy and repetition.

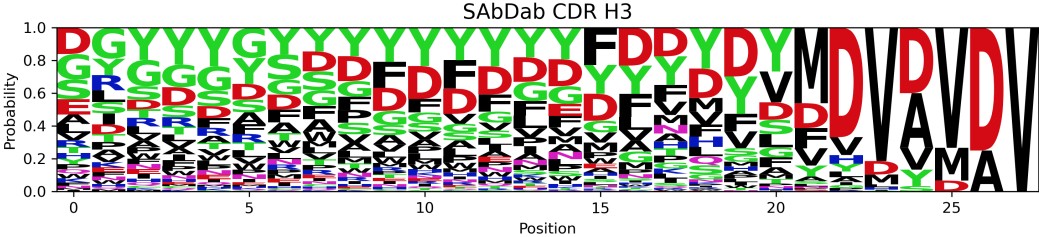

Figure 10: Distribution of CDR loop lengths across designs of specified heavy and light chain lengths. We note that HCDR length changes seem to directly correlate and possibly drive HCDR3 length, with smaller associations observed for the other chains.

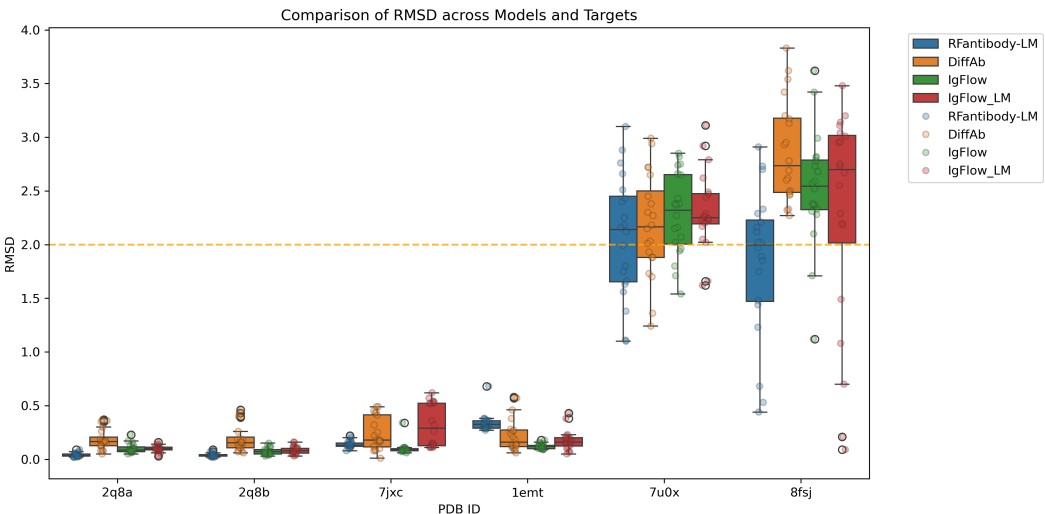

Figure 11: scRMSD distribution of RFantibody, DiffAb, IgFlow, and IgFlow-LM models when designing atypical CDR lengths in the CDR region.

