# OpenReview forum: "IgFlow-LM: De Novo Antibody Design via Joint Flow Matching on SE(3) and Protein Language Models Probability Flows"
_ICLR.cc/2026/Conference — Submitted to ICLR 2026_

### Official Review · Reviewer_pKAG · 2025-10-28

**Soundness:** 1
**Presentation:** 2
**Contribution:** 1
**Rating:** 2
**Confidence:** 4

**Summary:**

This paper proposed a flow-matching-based model for antibody sequence-structure codesign. It consists of a SE(3) equivariant flow matching module for structure generation and a latent flow matching module to generate protein language model embeddings which are then decoded into antibody sequences.

**Strengths:**

- The formulation of sequence-structure joint formulation is natural.
- Using pretrained protein language models might improve the sequence generation quality.

**Weaknesses:**

- This work overlooked the most of previously published research on generative models for antibody sequence-structure codesign [1,2,3,4,5]. It lacks both discussion and benchmarking.
- Using the flow matching technique to generate antibody has been explored a lot [1,2,3,5]. Combining diffusion models and protein language models to design antibodies has also been explored [2,6]. Therefore, the core contribution of this work is not novel.
- The evaluation of this work is mostly limited to comparing the distribution of the model and the datasets (RMSD, Ramachandran, LOGO Plot). There is no metrics that measure the property of the designed antibody (e.g. binding to specific antigens, humanness, etc).
- The conditional design setting (Section 4.2) is confusing. The section first says the generation is conditioned on the regions other than CDRs, but later it shows DockQ scores. DockQ scores assess the docking quality, so what target is used for docking? Is the antigenic target used as a condition? If so, what is the protocol for docking?
- Figure 4 and 5 do not convey useful information. Figure 4 shows that tyrosine (Y) dominates the generated antibody CDRs, which is an evidence of the model learning only trivial information. In Figure 5, the data points from Sabdab are smeared. There are no informative patterns shown in the figure.

[1] Antigen-Specific Antibody Design and Optimization with Diffusion-Based Generative Models for Protein Structures. 2022

[2] A Hierarchical Training Paradigm for Antibody Structure-sequence Co-design. 2023

[3] Atomically accurate de novo design of antibodies with RFdiffusion. 2024

[4] GeoAB: Towards Realistic Antibody Design and Reliable Affinity Maturation. 2024

[5] dyAb: Flow Matching for Flexible Antibody Design with AlphaFold-driven Pre-binding Antigen. 2025

[6] Antibody Design Using a Score-based Diffusion Model Guided by Evolutionary, Physical and Geometric Constraints. 2024

**Questions:**

See weaknesses.

---

> ### Author Response · Authors · 2025-12-04
> **Thank you sincerely for your thoughtful summary and encouraging feedback on our manuscript.**
>
> Thank you very much for your thorough summary and positive feedback on our manuscript. Your comprehensive comments and recognition of our work are precisely the motivation that drives us to keep improving. Please note that, due to technical limitations preventing image insertion, all figures referenced in the responses below can be found in the “pKAG” folder of the Supplementary Materials.
>
> Response to Question 1:
> We sincerely appreciate the reviewer’s comprehensive review and valuable suggestions. We have added baseline models that perform the same tasks as our IgFlow-LM.
>
> Response to Question 2:
> We sincerely appreciate the reviewer’s comprehensive review. We acknowledge that there are indeed several existing approaches that incorporate protein language models (PLMs) into structure-based models. However, these existing methods primarily use PLM embeddings as static inputs to provide prior knowledge [1, 2], and the PLM embeddings remain fixed throughout the generation process—they are not updated jointly with the structure. In contrast, our IgFlow-LM is a true end-to-end co-generation model that jointly optimizes both PLM embeddings and 3D structure. Moreover, our experiments demonstrate that this approach yields more stable performance compared to previous formulations.
>
> Response to Question 3:
> We sincerely appreciate the reviewer’s comprehensive review and valuable suggestions. First, we would like to clarify that in this study, the conditional CDR design task does not include antigen information. After careful discussion, we realized that our previously included antigen-binding experiment was actually inappropriate—since the model designs CDR regions without any antigen information, such an experiment could easily introduce binding bias. We acknowledge this as our oversight and bias. Therefore, we have removed this experiment and instead incorporated antibody humanization scores and ΔG of the heavy-light chain interface. These two evaluations are entirely independent of antigen information and can effectively assess the model’s ability to design antibody structures. Regarding your suggestion to use a larger dataset in the conditional CDR design task, we have increased the number of reference structures from 20 to 100. These 100 antibody structures cover common antibody variable region lengths and include non-canonical CDR lengths. The final experimental results are included in our revised manuscript.
>
> Response to Question 4:
> See Response to Question 3.
>
> Response to Question 5:
> We sincerely appreciate the reviewer’s comprehensive review and valuable suggestions. We acknowledge that, in general, such amino acid enrichment patterns could indeed indicate model bias. However, after analyzing the sequence logo of HCDR3 regions from natural antibodies in SAbDab (as shown in Figure R1), we found that natural antibodies also exhibit a certain degree of tyrosine (Y) enrichment. Moreover, the sequence logo distribution of IgFlow-LM closely resembles that of SAbDab. Therefore, this phenomenon is actually a bias introduced by the dataset itself. To further demonstrate IgFlow-LM’s ability to learn antibody sequence patterns, we additionally computed the amino acid frequency distributions for paired antibody sequences from OAS and compared them with those generated by IgFlow-LM (as shown in Figures R2, R3, and R4). The results show a high degree of similarity between the two. Regarding the potential impact of Y overexpression on antibody functionality, since our model does not design antibodies based on antigens, this question can be addressed in our future work. Concerning the embedding visualization, we have separated the two types of embedding points, and the results are shown in the revised manuscript.
>
> Additionally, we performed statistical analysis in the conditional CDR loop design task, and the results demonstrate that IgFlow-LM exhibits high robustness across different folding backends. In contrast, other generative models (IgFlow, DiffAb, RFantibody) show multiple statistically significant pairwise differences and greater variability across folding models, indicating lower consistency in structural fidelity.
>
> Reference
>
> [1]A Hierarchical Training Paradigm for Antibody Structure-sequence Co-design. 2023
>
> [2]Antibody Design Using a Score-based Diffusion Model Guided by Evolutionary, Physical and Geometric Constraints. 2024

---

### Official Review · Reviewer_W9wW · 2025-10-29

**Soundness:** 3
**Presentation:** 3
**Contribution:** 2
**Rating:** 4
**Confidence:** 3

**Summary:**

For sequence-structure co-design for de novo antibody, the authors proposed a new IgFlow-LM method that improves the existing IgFlow method by using continuous numerical representations with PLM instead of the discrete flow matching in IgFlow. Experiments showed higher diversity in the generated samples. However, continuous numerical representation PLM has already been used in the existing method, the novelty in terms of methodology seems not significant. I am also not convinced that the generated samples with higher diversity are realistic. Other than diversity, the improvements compared with IgFlow shown in Fig 3 and other tables and figures seem not significant.

**Strengths:**

For sequence-structure co-design for de novo antibody, the authors proposed a new IgFlow-LM method that improves the existing IgFlow method by using continuous numerical representations with PLM instead of the discrete flow matching in IgFlow. Experiments showed higher diversity in the generated samples.

**Weaknesses:**

1.	As presented in appendix A.1, continuous numerical representation PLM has already been used in the existing method (Ding et al 2019), the novelty in terms of methodology seems not significant.
2.	I am not convinced that the generated samples with higher diversity are realistic.
3.	Other than diversity, the improvements compared with IgFlow shown in Fig 3 and other tables and figures seem not significant.

**Questions:**

1.	As presented in appendix A.1, continuous numerical representation PLM has already been used in the existing method (Ding et al 2019), the novelty in terms of methodology seem not significant.
2.	I am not convinced that the generated samples with higher diversity are realistic.
3.	Other than diversity, the improvements compared with IgFlow shown in Fig 3 and other tables and figures seem not significant.
4.	The reference of IgFlow is never given.
5.	The meaning of SO(3) and SE(3) should be given. Not all the readers are familiar with these notations.

---

> ### Author Response · Authors · 2025-12-04
> **Thank you sincerely for your thoughtful summary and encouraging feedback on our manuscript.**
>
> Thank you very much for your thorough summary and positive feedback on our manuscript. Your comprehensive comments and recognition of our work are precisely the motivation that drives us to keep improving. Please note that, due to technical limitations preventing image insertion, all figures referenced in the responses below can be found in the “W9wW” folder of the Supplementary Materials.
>
> Response to Question 1:
> We sincerely appreciate the reviewer’s comprehensive review. We acknowledge that there are indeed several existing approaches that incorporate protein language models (PLMs) into structure-based models. However, these existing methods primarily use PLM embeddings as static inputs to provide prior knowledge [1, 2], and the PLM embeddings remain fixed throughout the generation process—they are not updated jointly with the structure. In contrast, our IgFlow-LM is a true end-to-end co-generation model that jointly optimizes both PLM embeddings and 3D structure. Moreover, our experiments demonstrate that this approach yields more stable performance compared to previous formulations.
>
> Response to Question 2:
> We sincerely appreciate the reviewer’s comprehensive review. We would like to elaborate on why the more diverse samples generated by our IgFlow-LM are authentic. First, as shown in Figure R1, the backbone dihedral angles of antibodies generated by IgFlow-LM align more closely with those of real antibodies than those from baseline models. Second, as illustrated in Figures R2, R3, and R4, the antibody sequences generated by IgFlow-LM better match the distributions observed in SAbDab (Figure R5) and OAS. Additionally, we performed structural relaxation using PyRosetta and found that IgFlow-LM exhibits the smallest structural deviation before and after relaxation, indicating that IgFlow-LM preserves the authenticity of generated antibodies while maintaining high fidelity.
>
> Response to Question 3:
> We sincerely appreciate the reviewer’s comprehensive review. We identified an error in our previous experimental setup and have therefore rerun all experiments. Furthermore, we have incorporated additional new experiments. Collectively, all results confirm that IgFlow-LM achieves the best overall performance, thanks to the synergistic modeling with the PLM.
>
> Response to Question 4:
> We sincerely appreciate the reviewer’s comprehensive review. We have added the relevant citations.
>
> Response to Question 5:
> We sincerely appreciate the reviewer’s comprehensive review. We have included the relevant description in the Supplementary Materials.
>
> Additionally, we performed statistical analysis in the conditional CDR loop design task, and the results demonstrate that IgFlow-LM exhibits high robustness across different folding backends. In contrast, other generative models (IgFlow, DiffAb, RFantibody) show multiple statistically significant pairwise differences and greater variability across folding models, indicating lower consistency in structural fidelity.
>
> Reference
>
> [1]A Hierarchical Training Paradigm for Antibody Structure-sequence Co-design. 2023
>
> [2]Antibody Design Using a Score-based Diffusion Model Guided by Evolutionary, Physical and Geometric Constraints. 2024

---

### Official Review · Reviewer_mKqa · 2025-10-30

**Soundness:** 2
**Presentation:** 2
**Contribution:** 3
**Rating:** 2
**Confidence:** 4

**Summary:**

The paper presents IgFlow-LM, a novel multi-modal generative model for de novo antibody design. The core contribution is a flow-matching framework that jointly generates the 3D backbone structure of an antibody and its corresponding sequence representation in the continuous latent space of a protein language model (PLM). Specifically, it combines an SE(3)-equivariant flow for the atomic coordinates with a probability flow over the PLM embeddings. The authors claim this joint, continuous approach avoids the limitations of methods that operate on discrete amino acid sequences, leading to generated antibodies with higher sequence diversity, improved structural fidelity, and stronger adherence to biophysical constraints. The authors demonstrate the effectiveness of the mode with two design tasks.

**Strengths:**

1. The idea of performing joint flow matching on both the SE(3) manifold for structure and the PLM embedding is simple and an elegant solution to the challenge of co-designing sequence and structure.

2. Experiments demonstrate the efficacy of the proposed method.

**Weaknesses:**

1. The innovation is limited. Adding PLM into a structure-based model is common in many protein-related models.

2. The primary baseline is IgFlow for ablation. The paper would be strengthened by more comprehensive comparisons against other state-of-the-art antibody co-design methods mentioned in the related works. In the evaluation, no sequence diversity is presented.

3. Some descriptions in the paper are not very clear. For example, the model IgFlow is not clearly defined. In table 1, the caption says RFDiffusion, but IgFlow in the table. And the evaluation of DockQ is not clear to me. In lines 407-408, is the framework aligned?

**Questions:**

See Weaknesses

---

> ### Author Response · Authors · 2025-12-04
> **Thank you sincerely for your thoughtful summary and encouraging feedback on our manuscript.**
>
> Thank you very much for your thorough summary and positive feedback on our manuscript. Your comprehensive comments and recognition of our work are precisely the motivation that drives us to keep improving.
>
> Response to Question 1:
> We sincerely appreciate the reviewer’s comprehensive review. We acknowledge that there are indeed several existing approaches that incorporate protein language models (PLMs) into structure-based models. However, these existing methods primarily use PLM embeddings as static inputs to provide prior knowledge [1, 2], and the PLM embeddings remain fixed throughout the generation process—they are not updated jointly with the structure. In contrast, our IgFlow-LM is a true end-to-end co-generation model that simultaneously and jointly optimizes both PLM embeddings and 3D structure. Moreover, our experiments demonstrate that this approach yields more stable performance compared to previous formulations.
>
> Response to Question 2:
> We sincerely appreciate the reviewer’s comprehensive review and valuable suggestions. We acknowledge that the baseline models used in our original study were indeed insufficient. Accordingly, in the unconditional design task, we have additionally included IgDiff as a new baseline model. For the conditional CDR design task, since our IgFlow-LM performs antigen-independent conditional CDR design, we have added RFantibody and DiffAb—both capable of antigen-independent conditional CDR design—as additional baseline models. Furthermore, in both design tasks, we have incorporated additional evaluations, including antibody humanization scores and ΔG of the heavy-light chain interface. The final experimental results have been incorporated into our revised manuscript.
>
> Response to Question 3:
> We sincerely appreciate the reviewer’s comprehensive review and valuable suggestions. We have corrected the errors in the manuscript. Regarding the DockQ evaluation, after careful discussion, we realized that our previously included antigen-binding experiment was actually inappropriate—since the model designs CDR regions without any antigen information, such an experiment could easily introduce binding bias. We acknowledge this as our oversight and bias. Therefore, we have removed this experiment and instead incorporated antibody humanization scores and ΔG of the heavy-light chain interface. These two evaluations are entirely independent of antigen information and can effectively assess the model’s ability to design antibody structures.
>
> Additionally, we performed statistical analysis in the conditional CDR loop design task, and the results demonstrate that IgFlow-LM exhibits high robustness across different folding backends. In contrast, other generative models (IgFlow, DiffAb, RFantibody) show multiple statistically significant pairwise differences and greater variability across folding models, indicating lower consistency in structural fidelity.
>
> Reference
>
> [1]A Hierarchical Training Paradigm for Antibody Structure-sequence Co-design. 2023
>
> [2]Antibody Design Using a Score-based Diffusion Model Guided by Evolutionary, Physical and Geometric Constraints. 2024

---

### Official Review · Reviewer_ABh9 · 2025-11-11

**Soundness:** 2
**Presentation:** 2
**Contribution:** 2
**Rating:** 2
**Confidence:** 5

**Summary:**

The paper proposes IgFlow-LM, a multimodal generative model for de novo antibody design that unifies SE(3)-equivariant structural flow matching (inspired by FrameFlow) with continuous flow matching in the latent space of a protein language model (PLM)—specifically IgBERT. The method jointly generates 3D backbone structures and PLM embeddings via a shared ODE-based flow-matching framework, followed by a decoder that maps embeddings back to discrete amino acid sequences. The authors evaluate IgFlow-LM on two tasks to validate the effectiveness of the proposed technique.

**Strengths:**

1. The integration of PLM latent flows with SE(3) structural flows is conceptually elegant and aligns with recent trends in multimodal protein generative modeling.

2. The paper demonstrates empirical results on structural validity: lower bond/angle deviations, better Ramachandran plot adherence (lower KL divergence), and high self-consistency across multiple folding predictors (ABB2, IgFold, ESMFold).

3. The use of germline-split data partitioning mitigates data leakage, enhancing result credibility.

**Weaknesses:**

1.	Lack of comparison with state-of-the-art diffusion-based PLM-integrated methods: In recent years, numerous works have successfully combined protein language models with diffusion frameworks for antibody design, including AbX, IgGM, and other Model such as DiffAb, dyMEAN, and RFantibody (an antibody-specific adaptation of RFdiffusion. These methods also perform joint or conditional sequence-structure co-design and report strong results on CDR generation, diversity, and structural fidelity. Without head-to-head evaluation on identical metrics and datasets, the performance advantage remains unsubstantiated.

2. Lack of functional validation: Despite claims about improved antigen binding (via DockQ), the docking experiment is based on only 20 antibodies, uses side-chain repacking only on CDRs, and reports high variance (DockQ: 0.47±0.18). Without binding affinity prediction (e.g., via deep learning or Rosetta) or wet-lab validation, functional superiority remains speculative.

3. Computational cost and scalability unaddressed: Flow matching with ODE integration is expensive. The paper does not report generation time, GPU memory usage, or scalability to full-length antibodies (only variable domains). This raises questions about practical utility.

**Questions:**

1. Comprehensive baseline comparison: Could the authors include comparisons with AbX, IgGM, DiffAb, dyMEAN, and RFantibody on the same conditional CDR design tasks using identical metrics (e.g., scRMSD, positional entropy, DockQ, developability scores)? This is essential to establish whether IgFlow-LM offers a genuine advance over current SOTA.

2. Functional relevance: Can the authors provide more robust evidence of improved antigen binding? For example, could they compute binding energy (ΔG) using Rosetta or a deep learning predictor (e.g., ABlooper, DeepAAntibody) on a larger set (≥100 antibodies)?

3. True co-design?: Is the PLM embedding updated during structure generation, or is it fixed from the ground-truth sequence? If fixed, doesn’t this mean the model is essentially doing structure-conditioned sequence generation, not joint co-design?

4.Generalization and robustness: How does IgFlow-LM perform on out-of-distribution antigens or non-canonical CDR lengths? The current test set is derived from SAbDab, which may not reflect therapeutic design scenarios.

5. Efficiency vs. benefit: Please report generation time per antibody and GPU memory consumption. Given the computational overhead of ODE integration, is the marginal gain in diversity/structure worth the cost compared to faster diffusion or inverse folding pipelines?

6. Tyrosine overrepresentation in HCDR3: In Figures 4, the generated HCDR3 sequences show an extreme enrichment of tyrosine (Y), especially at central positions. Is this reflective of the training data, or an artifact of the joint flow-matching objective or decoder bias? Could the authors compare the per-position amino acid frequencies in generated vs. natural HCDR3 (e.g., from SAbDab) to quantify this discrepancy? If Y is overrepresented, does this compromise the model’s capacity to generate non-canonical but functional binders (e.g., those rich in serine, glycine, or charged residues)?

---

> ### Author Response · Authors · 2025-12-04
> **Thank you sincerely for your thoughtful summary and encouraging feedback on our manuscript.**
>
> Thank you very much for your thorough summary and positive feedback on our manuscript. Your comprehensive comments and recognition of our work are precisely the motivation that drives us to keep improving. Please note that, due to technical limitations preventing image insertion, all figures referenced in the responses below can be found in the “ABh9” folder of the Supplementary Materials.
>
> Response to Weaknesses #1:
> We sincerely appreciate the reviewer’s comprehensive review and valuable suggestions. We acknowledge that the baseline models used in our original study were indeed insufficient. Accordingly, in the unconditional design task, we have additionally included IgDiff as a new baseline model. For the conditional CDR design task, since our IgFlow-LM performs antigen-independent conditional CDR design, we have added RFantibody and DiffAb—both capable of antigen-independent conditional CDR design—as additional baseline models. Furthermore, in both design tasks, we have incorporated additional evaluations, including antibody humanization scores and ΔG of the heavy-light chain interface. The final experimental results have been incorporated into our revised manuscript.
>
> Response to Weaknesses #2:
> We sincerely appreciate the reviewer’s comprehensive review and valuable suggestions. First, we would like to clarify that in this study, the conditional CDR design task does not include antigen information. After careful discussion, we realized that our previously included antigen-binding experiment was actually inappropriate—since the model designs CDR regions without any antigen information, such an experiment could easily introduce binding bias. We acknowledge this as our oversight and bias. Therefore, we have removed this experiment and instead incorporated antibody humanization scores and ΔG of the heavy-light chain interface. These two evaluations are entirely independent of antigen information and can effectively assess the model’s ability to design antibody structures. Regarding your suggestion to use a larger dataset in the conditional CDR design task, we have increased the number of reference structures from 20 to 100. These 100 antibody structures cover common antibody variable region lengths and include non-canonical CDR lengths. The final experimental results are included in our revised manuscript.
>
> Response to Weaknesses #3:
> See response to Question 5.
>
> Response to Question 1:
> We sincerely appreciate the reviewer’s comprehensive review and valuable suggestions. In conditional CDR region design, since our IgFlow-LM performs antigen-independent conditional CDR design, we have added RFantibody and DiffAb—both capable of antigen-independent conditional CDR design—as baseline models. The final experimental results are included in our revised manuscript.
>
> Response to Question 2:
> See response to Weaknesses #2.
>
> Response to Question 3:
> We sincerely appreciate the reviewer’s comprehensive review. During unconditional design with IgFlow-LM, both PLM embeddings and structure are updated. In conditional CDR region design, the PLM embeddings of the CDR regions are replaced by initial noise, while those of other regions remain unchanged.
>
> Response to Question 4:
> We sincerely appreciate the reviewer’s comprehensive review and valuable suggestions. We plotted the scRMSD distributions of four models when designing CDR regions with non-canonical CDR lengths, as shown in Figure R1. The HCDR3 lengths of these six antibodies (PDB IDs) are 3, 3, 4, 5, 20, and 25, respectively. These HCDR3 lengths are relatively rare in SAbDab and thus represent non-canonical CDR lengths. As shown, all models perform well on short CDRs but encounter difficulties with long CDRs. Our IgFlow-LM, however, consistently performs reasonably well across non-canonical CDR lengths. Regarding your comment on the dataset, we acknowledge that using only SAbDab may not be suitable for therapeutic applications. However, the primary goal of this study is to explore the potential of joint generation of antibody structures and PLM embeddings; exploration of other datasets is left for future work, as mentioned in our Conclusion.
>
> Response to Question 5:
> We sincerely appreciate the reviewer’s comprehensive review and valuable suggestions. We acknowledge that computational resource requirements during practical deployment can significantly impact real-world applicability. Therefore, we have additionally compared the inference times required by all baseline models across both tasks, as shown in Figure R6. All experiments were conducted on one A100 PCIE 40G GPU, performing 100 sampling runs and averaging the time per sample. As shown, IgFlow-LM achieves superior performance compared to IgFlow with only a modest increase in inference time, and its inference time remains relatively low compared to other models, making it suitable for large-scale biomolecular prediction tasks.This table has been added to the manuscript.

---

> ### Author Response · Authors · 2025-12-04
>
> Response to Question 6:
> We sincerely appreciate the reviewer’s comprehensive review. We acknowledge that, in general, such amino acid enrichment patterns could indeed indicate model bias. However, after analyzing the sequence logo of HCDR3 regions from natural antibodies in SAbDab (as shown in Figure R2), we found that natural antibodies also exhibit a certain degree of tyrosine (Y) enrichment. Moreover, the sequence logo distribution of IgFlow-LM closely resembles that of SAbDab. Therefore, this phenomenon is actually a bias introduced by the dataset itself. To further demonstrate IgFlow-LM’s ability to learn antibody sequence patterns, we additionally computed the amino acid frequency distributions for paired antibody sequences from OAS and compared them with those generated by IgFlow-LM (as shown in Figures R3, R4, and R5). The results show a high degree of similarity between the two. Regarding the potential impact of Y overexpression on antibody functionality, since our model does not design antibodies based on antigens, this question can be addressed in our future work.
>
> Additionally, we performed statistical analysis in the conditional CDR loop design task, and the results demonstrate that IgFlow-LM exhibits high robustness across different folding backends. In contrast, other generative models (IgFlow, DiffAb, RFantibody) show multiple statistically significant pairwise differences and greater variability across folding models, indicating lower consistency in structural fidelity.

---

### Author Response · Authors · 2025-12-04
**Brief introduction to our submission**

Dear Area Chair,

Thank you very much for handling our submission.
We would like to briefly highlight the key contributions and context of our paper:
Paper title: IGFLOW-LM: De Novo Antibody Design via Joint Flow Matching on SE(3) and Protein Language Model Probability Flows
Summary:
Our work presents IgFlow-LM, a multi-modal generative framework that jointly models SE(3)-equivariant structural flows and PLM-based probabilistic flows for antibody design. By extending FrameFlow to incorporate PLM latent embedding generation, our method enables consistent co-design of antibody 3D structures and sequences.
Key contributions:
A new multimodal flow-matching framework that unifies SE(3) structural generation and continuous PLM latent flows.
Joint sequence–structure co-design that improves physical plausibility and structural consistency.
Extensive experiments showing that IgFlow-LM:
Outperforms IgFlow, DiffAb, RFantibody on both unconditional and CDR-conditioned design tasks.
Produces antibodies with better physical geometry, higher sequence diversity, and improved humaneness.
Demonstrates strong cross-folding-backend robustness (ABB2, IgFold, ESMFold).
We hope these clarifications are helpful while you assess the submission.
Thank you again for your time and effort.

Best regards,
Authors

---

### Meta-Review · Area_Chair_4tcJ · 2026-01-06

**Summary:**

The main concern of this paper is the lack of comparison with state-of-the-art diffusion-based PLM-integrated methods. All reviewers have voted for rejecting this paper.

**Reviewer Concerns:**

The reviewer concerns are not addressed

**Reviewer Scores:**

The reviewers would have kept their score even if they had participated fully in the discussion.

---

### Decision · Program_Chairs · 2026-01-26

Reject